# Efficacy of Agricultural and Food Wastes as the Growing Media for Sunflower and Water Spinach Microgreens Production

**Ornprapa Thepsilvisut [1,*], Nipawadee Sukree [1], Preuk Chutimanukul [1], Dusit Athinuwat [1], Wilawan Chuaboon [1], Phakpen Poomipan [1], Vorapat Vachirayagorn [1], Nuttaporn Pimpha [2], Panita Chutimanukul [3] and Hiroshi Ehara [4]**

1   Department of Agricultural Technology, Faculty of Science and Technology, Thammasat University Rangsit Center, Klong Luang 12120, Pathum Thani, Thailand; nipawadee.su@gmail.com (N.S.); plove9528@hotmail.com (P.C.); athinova6@hotmail.com (D.A.); wchuaboon@gmail.com (W.C.); phakpen@tu.ac.th (P.P.); vore405@me.com (V.V.)

2   National Nanotechnology Center (NANOTEC), 111 Thailand Science Park, Klong Luang 12120, Pathum Thani, Thailand; nuttaporn@nanotec.or.th

3   National Center for Genetic Engineering and Biotechnology (BIOTEC), National Science and Technology Development Agency, Klong Luang 12120, Pathum Thani, Thailand; panita.chu@biotec.or.th

4   International Center for Research and Education in Agriculture, Graduate School of Bioagricultural Sciences, Nagoya University, Nagoya 464-8601, Japan; ehara@agr.nagoya-u.ac.jp

*   Correspondence: ornprapa@staff.tu.ac.th

**Abstract:** The growing media is one of the significant elements affecting microgreens' yield and quality. This experiment investigated the possibility of waste utilization instead of employing peat moss to produce sunflower and water-spinach microgreens. The treatments consisted of peat moss (Control), coconut coir dust (CD), leaf compost (LC), food waste compost (FC), CD:LC = 1:1 $v/v$, CD:FC = 1:1 $v/v$, LC:FC = 1:1 $v/v$, and CD:LC:FC = 1:1:1 $v/v$. The results proved that the highest yield of sunflower microgreens was observed when cultivated in 1:1 $v/v$ of CD:LC media (10,114.81 g m$^{-2}$), whereas the highest yield of water spinach microgreens was recorded under the treatments of CD, Control, 1:1 $v/v$ of CD:LC, and 1:1:1 $v/v$ of CD:LC:FC media (10,966.67–9800.00 g m$^{-2}$). The biochemical composition of the microgreens varied within the types. Our findings demonstrated that a tendency of an increase in chlorophyll and carotenoid contents depended on the growth of both microgreens under different growing media. All growing media did not cause excess nitrate residue or pathogenic contamination in both microgreens, namely *Clostridium perfringens*, *Salmonella* spp., and *Staphylococcus aureus*. In contrast, almost all the growing media resulted in a higher population of *Bacillus cereus* contamination in both microgreens than the standard set limit, except for sunflower microgreens grown in the control and CD growing media. These findings could suggest that the 1:1 $v/v$ of CD:LC and CD media were the most effective growing media for sunflower and water spinach microgreens, respectively, but further cleaning before consumption is recommended to avoid or reduce the foodborne incidences caused by *B. cereus* in microgreens.

**Keywords:** food waste; agricultural waste; microgreens; germination index; yield quality

## 1. Introduction

Microgreens, edible immature greens, are generally obtained from different seed species, including vegetables, cereals, herbs, flowers, or edible wild plants [1–4]. The harvest cycles of microgreens vary from 7–28 days after germination, when two cotyledon leaves are completely expanded with or without the emergence of the first true leaves, and the height is usually 5–10 cm [1,5]. Microgreens are considered an alternative vegetable or functional food for consumers who are aware of their health and antioxidant aging benefits due to their high nutraceutical values, such as phytochemicals (e.g., polyphenols, carotenoids, flavonoids), vitamins, and mineral nutrients [6–8], in addition to the several advantages of various colors, flavors, and textures. Microgreens have recently gained

popularity; they are commonly consumed fresh in mixed salads or as supplements added to healthy drinks or as a garnish.

Microgreens can be readily grown irrespective of the season in a greenhouse or indoors with artificial lighting systems and can produce multiple cycles compared to mature ones. Moreover, they are often grown in hydroponic systems with a nutritious liquid solution or a thin layer of different growing media mixed with solid fertilizer, depending on the microgreen variety and the production scale [9,10]. A peat-based growing media is typically used for microgreen production because of its appropriate water-retention properties and good aeration. However, peat moss has to be imported from European countries, which hence becomes quite expensive, and it is also considered a non-renewable resource because its growth rate is extremely slow and takes centuries to fully develop [11]. Nevertheless, there are some studies to demonstrate and find an alternative substrate to peat moss, for example, textile-fiber mat [12], coconut fiber, jute fabric [13], and coconut coir dust mixed with peat or sugarcane filter cake [11]. Di Gioia et al. [14] noted that favorable growing media should contain micropores holding water and macropores to allow the excess water to drain away and prevent waterlogging, which could promote better root development. Furthermore, the media should have the appropriate pH (in the range of 5.5–6.5) and be microbiologically safe. The growing media is one of the reasonable risk factors for microgreen safety as it has been reported that *Escherichia coli* could be transferred from peat moss or perlite to the edible part of microgreens [15]. In addition to the environmental sustainability of the microgreen production process, the ideal growing media should be relatively inexpensive, locally available, and derived from renewable sources [14].

Recently, home-grown microgreens or their production on a local farm scale is increasingly popular to ensure immediate consumption after harvest due to their rapid quality deterioration caused by high respiration rates and delicate leaves, depending upon the species [5,16]. To support local-scale microgreen production, finding suitable growing media made from household or agricultural wastes (e.g., food waste, leaf litter, sawdust, livestock waste) is needed. The utilization of these organic wastes also has a positive impact on the environment. Several studies investigate food waste or recycling waste from different sources to produce compost for growing vegetables containing many plant nutrients [17–19]. However, the excessive salt content is the most common obstacle to a food-waste compost, which has to be used in optimal doses or mixed with other ingredients of the growing media [18]. However, the growing media serves as one of the main costs for microgreen production and plays a significant role in affecting yield and phytochemical composition, as well as being one of the microbial contamination risks of the microgreens. The basic scientific data are not widely published. The result of this study could be valuable for offering a safe and cost-effective growing media for use in sustainable microgreen production, especially for local farms or small-scale family businesses, further increasing the added value of microgreen products.

## 2. Materials and Methods

### 2.1. Experimental Designs and Growing Media Preparation

The experiment was arranged in a completely randomized design (CRD) with three replicates. The treatments were eight different growing media consisting of T1: peat moss (Control); T2: coconut coir dust (CD); T3: leaf compost (LC); T4: food waste compost (FC); T5: CD:LC = 1:1 *v/v*; T6: CD:FC = 1:1 *v/v*; T7: LC:FC = 1:1 *v/v*; T8: CD:LC:FC = 1:1:1 *v/v*. The commercial peat moss and coconut coir dust were purchased from the agriculture-equipment market in Pathum Thani province, Thailand (Latitude: 14.08308, Longitude: 100.63473). The following procedure was adopted from the methodology described in Popradit et al. [20] for leaf compost preparation. The leaf wastes (dry leaves) and cow dung were collected from the agricultural farm at Thammasat University, Thailand (Latitude: 14.07437, Longitude: 100.60921). The leaf wastes were placed in thin layers (less than 10 cm) into the pile, alternating with cow dung according to the ratio of 4:1 ratio by volume. The leaf compost pile was stacked in a triangular shape with a height higher than 1.50 m and a

base width of 2.5 m. The humidity in the pile was controlled at 60–70% moisture content to enhance the microbial activity in the decomposition process, and an adequate amount of water was added when the compost pile was found dry. The leaf compost pile was left for 60 days without turning over. The compost was finished when the pile was no longer heating up and the original materials turned earthy and black. The compost was dried in a shading room, collected to sieve, and kept in a plastic bag.

For the preparation of food-waste compost, the types of food waste were separated into three groups consisting of (1) vegetables and fruit peels, (2) leftover foods and rice (with wastewater drained), and (3) bones and seafood shells, which were collected from the kitchen and student dining areas in the university. The proportion of all types of food waste used in this experiment was approximately 5:4:1 ratio by weight, respectively. Dry leaves were used as a supplementary material combined with the mixed food waste at a 1:4 ratio by weight to provide minimum aeration, absorb moisture, and ensure the correct C/N ratio proportion to reduce undesired odor [21]. All the waste materials were put into a 60 L food-waste bin and stirred every 3 days and kept to decompose under an aerobic condition for 60 days. The decomposed food waste was flattened, air-dried, sieved, and kept in shady conditions.

### 2.2. Plant Materials and Growing Conditions

Two microgreens of sunflower (*Helianthus annuus* L.) and water spinach (*Ipomoea aquatica* Forsk.), which are both commonly consumed in Thailand, were selected in this study. The seeds of these microgreens were surface-sterilized with a commercial NaOCl solution (200 mg $L^{-1}$) for 5 min under constant agitation, rinsed with drinking water for 1 min, and then soaked in drinking water for 6 h to overcome seed dormancy via seed imbibition to trigger the activation of various metabolic processes for germination [22]. To evaluate the agronomic performance of each media, the soaked seeds at 90 g of sunflower (500 g $m^{-2}$) and 180 g of water spinach (1000 g $m^{-2}$) were evenly broadcast on the surface of each growing media in a tray (30 cm $\times$ 60 cm $\times$ 3 cm) and covered up with a thin layer (0.5 cm) of the same growing media. After sowing, the trays were placed on a bench and arranged in a randomized complete design with three replications (trays). The sown trays were irrigated manually using a water nozzle and covered for two days with a second germination tray that had been decontaminated with 70% ethanol and inverted to form a lid to retain moisture and promote seed germination in near darkness at 23 $\pm$ 3 °C. After that, the two microgreens were grown under a controlled environment at 25 $\pm$ 3 °C and 60 $\pm$ 5% relative humidity (RH) along with a 16:8 h photoperiod using a cool daylight 6500 K (3070 lumens) fluorescent lamp in a greenhouse under different growing-media conditions. The microgreens were misted with sterile DI water two times a day based on the preliminary cultivation trials and visual inspection, whereby approximately 200 and 300 mL of sterile DI water was poured directly over the growing media, and the trays were gently rocked to distribute the water evenly. Ten days after sowing, the microgreens of each tray were harvested by cutting the seedlings just above the surface of the growing media with a sterilized knife and used for yield and quality-assessment analysis.

### 2.3. Data Collection and Analysis

2.3.1. Physicochemical Properties of Growing Media

As the physical properties, bulk density and total pore space were determined according to the European Standard 130411 (European Standard, 1999), following the modified method described by Di Gioia et al. [12]. The dry growing media (dried at 105 $\pm$ 1 °C) was transferred to knowing-volume cylinders. Then, the bulk density, defined as the dry mass of the growing media sample in a given volume, was calculated. In addition, particle density was determined using the conical flask as a pycnometer, and the total pore space was estimated from the bulk and particle densities.

To analyze the growing media's chemical properties before planting, each treatment sample was air-dried at room temperature, crushed, and sieved through a 2-mm mesh

sieve for further analysis. The pH and electrical conductivity (EC) of each growing media sample were measured in water at a ratio of 1:10, determined by using a pH-EC meter (SciberScanPC510, EUTEC, Singapore). The organic matter was quantified using the Walkley and Black [23] method, and the total N content was determined by using the Kjeldahl method. The concentrations of P, K, Mg, Ca, and Na in the growing media were determined according to the standard protocol of the Association of Official Analytical Chemists (AOAC) with modifications. Next, 1.0 g of each sample was digested in 15 mL of nitric-perchloric acid ($HNO_3$:$HClO_4$ at a ratio of 2:1 $v/v$). After digestion, the samples were diluted with distilled water until they reached a volume of 50 mL, and then they were stored in plastic tubes at room temperature. The amount of P in the distilled samples was analyzed by using a spectrophotometer (UV-1280, Shimadzu, Japan) at 420 nm. In contrast, the K, Mg, Ca, and Na quantifications were analyzed using an atomic absorption spectrometer (PinAAcle900F, Perkin-Elmer, Waltham, MA, USA).

### 2.3.2. Germination Test

All growing media treatments consisted of three replicates with 100 seeds in each. The sunflower seeds and water spinach were surface-sterilized with a commercial NaOCl solution (200 mg $L^{-1}$) for 5 min under constant agitation and then rinsed five times with drinking water. The sterilized seeds were sown in nursery trays filled with different growing media and left to germinate at a controlled temperature of $25 \pm 3$ °C. Irrigation was applied immediately after seed sowing and repeated daily until the final emergences. The number of germinated seeds for each plant was recorded daily for 10 days. The seeds were considered completely germinated once the root protruded about 2 mm through the pericarp [24]. Three germination parameters, including germination percentage, germination index, and mean germination times, were calculated by the following formulas:

$$\text{Germination rate} = [(\text{Number of germinated seeds}/\text{Number of seeds tested}) \times 100] \quad (1)$$

$$\text{Germination index} = \sum (\text{Number of germinated seeds}/\text{Days of the last count}) \quad (2)$$

$$\text{Mean Germination Time} = \sum dn / \sum n \quad (3)$$

where

n = the number of seeds which were germinated on day D,
D = the number of days counted from the beginning of germination [24].

### 2.3.3. Yield Assessment

The harvested microgreens were separated into marketable (the cotyledon leaves had developed with one set of true leaves, approximately 5–10 cm tall) and non-marketable microgreens (non-fully developed cotyledon leaves, less than 5 cm tall). The marketable and non-marketable microgreen groups were immediately weighed to determine the fresh weight of the shoots and the non-marketable yield per unit area (g m$^{-2}$). The waste-yield percentage was determined by calculating the ratio of non-marketable yield and total harvested microgreens per unit area and multiplying the result by 100. The dry matter was measured by oven-drying (Thermotec 2000, Contherm Scientific Ltd., Lower Hutt, New Zealand) at 60 °C until constant sample weights were achieved. The water content and dry matter of the marketable shoot yield of each microgreen were calculated as in the formulas shown below [25]:

$$\text{Shoot water content (\%)} = [(\text{fresh weight} - \text{dry weight})/\text{fresh weight}] \times 100 \quad (4)$$

$$\text{Shoot dry matter (\%)} = 100 - \text{shoot water content (\%)} \quad (5)$$

2.3.4. Phytochemical Analysis

The total chlorophyll and carotenoid contents were determined by adapting a technique described by Bulgari et al. [13]. The extracts were prepared by weighing 1.0 g of chopped fresh shoot samples and mixing them with 10 mL of 80% acetone. After incubation at 4 °C for 24 h in the dark, the supernatant was placed in a 1.0 cm pathlength cuvette, and the absorbance readings were measured at 645, 663, and 470 nm using an ultraviolet spectrophotometer (UV-1280, Shimadzu, Japan). The absorbances were used to calculate the contents of chlorophyll and carotenoid and expressed as mg 100 $g^{-1}$ fresh weight by using the following formulas [26,27]:

$$\text{Chlorophyll a (Chl a)} = [(12.7 \times A663) - (2.69 \times A645)] \times (V/1000\,W) \tag{6}$$

$$\text{Chlorophyll b (Chl b)} = [(22.9 \times A645) - (4.68 \times A663)] \times (V/1000\,W) \tag{7}$$

$$\text{Total chlorophyll} = [(20.20 \times A645) + (8.02 \times A663)] \times (V/1000\,W) \tag{8}$$

$$\text{Carotenoid} = [((1000 \times A470) - (3.27 \times \text{Chl a}) - (104 \times \text{Chl b}))/229] \times (V/1000\,W) \tag{9}$$

where

A663, A645, and A470 = the absorbances at 663, 645, and 470 nm, respectively;
V = the volume of the extraction solution;
W = the mass of the fresh shoot sample.

The total phenolic content and antioxidant activity were determined according to the method of Chutimanukul et al. [28], with modifications. Briefly, for the sample extraction, 10 mg of the dried sample was homogenized with 5 mL of methanol containing 1% hydrochloric acid at 25 ± 1 °C for 3 h and centrifuged at 12,000× $g$ for 5 min. The supernatants were collected, and the total phenolic content and antioxidant activity were evaluated. A 200 µL aliquot of the obtained sample was mixed with 200 µL of 1 N Folin–Ciocalteu reagent and 600 µL of 7.5% sodium carbonate solution to determine the total phenolic content. The mixture was incubated at 25 ± 1 °C for 1 h, and the absorbance was measured at 730 nm with an ultraviolet spectrophotometer. The total phenolic content was calculated from the gallic acid calibration curve, and the result was expressed as mg gallic acid equivalent (GAE) 100 $g^{-1}$ dry weight. The quantity of flavonoid content was measured by the colorimetric method. The aliquot sample of 350 µL was mixed with 75 µL of 5% sodium nitrite, 75 µL of 10% aluminum chloride, and 500 µL of 1 M sodium hydroxide, respectively. The mixture was vortexed and incubated at 25 ± 1 °C for 15 min. Subsequently, the mixture solution was centrifuged at 12,000× $g$ for 2 min, and the absorbance was read at 515 nm with an ultraviolet spectrophotometer. Flavonoid content was determined based on a rutin standard curve, and the result was expressed as mg rutin equivalents 100 $g^{-1}$ dry weight. To assay the 2,2-diphenyl-1-picrylhydrazyl (DPPH) radical scavenging, an aliquot of filtrate (100 µL) was mixed with 900 µL of 0.1 mM DPPH working solution and then incubated in the dark for 30 min at 25 ± 1 °C. The absorbance of the mixed solution was measured at 515 nm. A calibration curve was determined using trolox, and the result was expressed with the inhibition percentage of DPPH absorbance as in the following formulas:

$$\text{DPPH radical scavenging (\%)} = [(Ac - As)/Ac] \times 100 \tag{10}$$

where

Ac = control reaction absorbance; As = sample reaction absorbance

The nitrate content was measured according to the methodology described by Cataldo et al. [29], with modifications. One g fresh weight sample was ground in 5 mL of distilled water. The extract was centrifuged at 4000× $g$ for 15 min, and the supernatant was ad-

justed to 10 mL. After that, 0.1 mL of the obtained sample was used for the colorimetric determination of nitrate by using the brucine–sulfanilic acid method at 410 nm utilizing a spectrophotometer. The nitrate concentration was calculated from the calibration curves of the standards, and the sample weight was converted for reporting as mg kg$^{-1}$ fresh weight.

### 2.3.5. Microbiological Analysis

For microbiological analysis to evaluate the microbial contamination on microgreens grown on each growing media, the experiment was conducted in separated samples using the same procedures described in the growing condition. Three trays per growing media of each microgreen plant were arranged in a separate greenhouse to avoid cross-contamination, except for the growing media effect. All materials and equipment were carefully sprayed with 70% ethanol or washed with a commercial NaOCl solution (200 mg L$^{-1}$) before use. At the harvest stage (5–10 days after sowing), 100 g of microgreens were randomly selected from each tray and cut with a pair of sterilized scissors, and then washed with deionized water and packed into a sterile plastic bag for transporting to the laboratory for microbiological assessment.

The viable cell count was measured using a spread-plate culture method to investigate the effect of each growing media on pathogenic-bacteria contamination in microgreens. Plates with selective and nonselective media were incubated at 37 °C for 48 h. The number of colonies on each plate was then counted. Bacterial counts from duplicate plates were reported in CFU g$^{-1}$, following the standard methods of ISO 7932:2004, ISO 7937:2004, ISO 6579-1:2017, and AOAC 2003.07 for analysis of the presence of *Bacillus cereus*, *Clostridium perfringens*, *Salmonella* spp., and *Staphylococcus aureus*, respectively. In brief, for the detection of *B. cereus*, the serial ten-fold dilution samples were brought into a sterile petri dish containing mannitol-egg yolk polymyxin (MYP) agar medium using a spread-plate technique and then incubated at 37 °C for 24 h. After incubation, the presumptive colonies (large, pink, irregular margins) were counted on each plate and reported in CFU g$^{-1}$. To identify *C. perfringens* in microgreen samples, 1 mL of the initial suspension samples was poured into a sulfite-cycloserine (SC) agar medium using the poured-plate technique and then incubated at 37 °C for 20 h. After incubation, the plates containing more than 150 colonies were selected to count the presumptive colonies (medium size, grey, translucent) on each plate and reported in CFU g$^{-1}$. The initial suspension after enrichment of the samples in SC broth at 37 °C for 24 h was inoculated from the upper one-third of the broth onto the surface of the xylose lysine desoxycholate (XLD) agar and Bismuth sulfite (BS) agar and then incubated at 37 °C for 24 h for the detection of *Salmonella* spp. contamination in microgreen samples. After incubation, the presumptive colonies (green with little or no darkening of the surrounding medium for BS agar plates and red or black colonies for XLD agar plates) were counted on each plate and reported in CFU g$^{-1}$. Finally, to detect *S. aureus*, the serial ten-fold dilution samples were transferred into a sterile petri dish containing Baird–Parker medium (BPM) agar using a spread-plate technique and then incubated at 35–37 °C for 24 h. After incubation, the presumptive colonies (black or grey, convex, and opaque) were counted on each plate and reported in CFU g$^{-1}$.

### 2.4. Statistical Analysis

Experimental treatment effects were analyzed using a completely randomized design with three replications. Data were analyzed through one-way analysis of variance (ANOVA) by using IBM SPSS Statistics, Version 26.0 software (IBM Corp., Armonk, NY, USA). The mean treatment difference values were compared using Duncan's multiple range tests, with the significance determined at $p \leq 0.05$.

## 3. Results

### 3.1. Physicochemical Properties of Growing Media

The lowest bulk density and highest total pore space were observed in coconut coir dust media, with values of 67.64 kg m$^{-3}$ and 60.27%, respectively. Contrarily, 100% food-

waste compost media showed the significantly highest bulk density (594.26 kg m$^{-3}$) and lowest total pore space (14.05%); this total pore space had, nevertheless, no significant difference from the growing media consisting of 1:1 *v/v* of coconut coir dust + food waste compost (15.70%). However, it could be seen that 1:1 *v/v* of coconut coir dust + leaf compost showed a similar bulk density to the peat moss (control), which was 251.86 and 263.62 kg m$^{-3}$, respectively (Table 1). The different proportions of organic waste in the growing media significantly affected the chemical properties compared to the commercial peat moss as the control treatment. Coconut coir dust had the lowest pH (5.50), followed by commercial peat moss (pH 5.98), whereas the leaf compost, food-waste compost, and mixed-organic waste media (T5-T8) had pH values between 6.53 to 6.82. In terms of electrical conductivity (EC), the growing media consisting of food-waste compost, coconut coir dust, and the two mixed half-and-half (coconut coir dust: food-waste compost = 1:1 *v/v*) had significantly higher EC values (3.41–3.61 dS m$^{-1}$). Excluding the commercial peat moss, the organic matter in the coconut coir dust treatment was much higher than in the other growing media (Table 1).

**Table 1.** Selected physicochemical properties of different growing media for microgreens production.

| Treatment | Bulk Density (kg m$^{-3}$) | Total Pore Space (%) | pH (1:10 H$_2$O) | EC (1:10 H$_2$O) (dS m$^{-1}$) | Organic Matter (%) |
|---|---|---|---|---|---|
| T1 (Control) | 251.86 ± 9.40 f | 37.61 ± 1.84 c | 5.98 ± 0.03 b | 1.59 ± 0.06 d | 35.42 ± 5.83 b |
| T2 | 67.64 ± 3.64 e | 60.27 ± 0.74 a | 5.50 ± 0.44 c | 3.44 ± 0.19 a | 40.54 ± 1.05 a |
| T3 | 482.66 ± 4.57 c | 36.17 ± 4.33 c | 6.59 ± 0.11 a | 1.67 ± 0.10 d | 14.03 ± 1.60 cd |
| T4 | 594.26 ± 4.26 a | 14.05 ± 2.71 e | 6.77 ± 0.02 a | 3.61 ± 0.42 a | 10.06 ± 0.52 d |
| T5 | 263.62 ± 10.91 f | 46.80 ± 5.65 b | 6.53 ± 0.05 a | 2.09 ± 0.10 c | 17.66 ± 0.68 c |
| T6 | 347.96 ± 15.83 e | 15.70 ± 0.08 e | 6.70 ± 0.07 a | 3.41 ± 0.39 a | 11.69 ± 0.50 d |
| T7 | 543.63 ± 5.93 b | 22.97 ± 1.21 d | 6.82 ± 0.00 a | 2.43 ± 0.18 bc | 10.01 ± 2.21 d |
| T8 | 396.46 ± 2.82 d | 22.61 ± 7.10 d | 6.68 ± 0.07 a | 2.67 ± 0.16 b | 10.92 ± 1.61 d |
| *p*-value | 0.000 | 0.000 | 0.000 | 0.000 | 0.000 |

T1: commercial peat moss (Control); T2: coconut coir dust (CD); T3: leaf compost (LC); T4: food waste compost (FC), T5: CD:LC = 1:1 *v/v*, T6: CD:FC = 1:1 *v/v*, T7: LC:FC = 1:1 *v/v*, T8: CD:LC:FC = 1:1:1 *v/v*). According to Duncan's multiple range test at $p \leq 0.05$, the mean with different letters in the same column indicates a significant difference.

Nitrogen (N) in the coconut coir dust treatment was 2 times lower than in the control treatment. In contrast, the leaf compost, food-waste compost, and mixed-organic waste media (T5-T8) showed no difference or had significantly higher N than that of the control treatment, especially the treatments of leaf compost and 1:1 *v/v* of coconut coir dust: food waste compost (16.11 and 15.05 mg g$^{-1}$, respectively). Although the phosphorus (P) concentration in the coconut coir dust treatment was lower than in the control treatment, the other growing media treatments showed an increase in P concentration of nearly 3 times more than that of the control. In addition, the different growing media had potassium (K) concentrations ranging from 7.88 to 9.91 mg g$^{-1}$, which were significantly higher than the control treatment. The highest calcium (Ca) concentration was found for the treatment of food-waste compost (64.19 mg g$^{-1}$), whereas the highest magnesium (Mg) concentration was obtained for the treatment of leaf compost (5.65 mg g$^{-1}$). For the overall trend for salinity, the growing media consisting of 100% food-waste compost half-mixed with coconut coir dust (CD:FC = 1:1 *v/v*) and leaf compost (LC:FC = 1:1 *v/v*) had significantly higher Na concentrations ranging from 6.11 to 7.06 mg kg$^{-1}$, which had no significant difference from the control treatment, which was 5.46 mg kg$^{-1}$ (Table 2).

**Table 2.** Nutrient concentrations in different growing media for microgreens production.

| Treatment | N (mg g$^{-1}$) | P (mg g$^{-1}$) | K (mg g$^{-1}$) | Ca (mg g$^{-1}$) | Mg (mg g$^{-1}$) | Na (mg kg$^{-1}$) |
|---|---|---|---|---|---|---|
| T1 (Control) | 13.60 ± 0.10 cd | 12.45 ± 3.04 e | 3.65 ± 0.22 b | 57.15 ± 1.52 b | 1.99 ± 0.00 g | 5.46 ± 0.58 abc |
| T2 | 5.99 ± 0.36 e | 9.68 ± 3.59 e | 9.91 ± 1.16 a | 19.37 ± 1.72 f | 1.49 ± 0.00 h | 4.64 ± 1.31 bcd |
| T3 | 16.11 ± 0.16 a | 53.65 ± 6.05 a | 8.46 ± 0.33 a | 34.37 ± 3.95 e | 5.65 ± 0.01 a | 3.58 ± 0.55 d |
| T4 | 14.41 ± 1.78 bc | 38.45 ± 2.99 d | 7.56 ± 1.00 a | 64.19 ± 1.79 a | 5.31 ± 0.01 d | 7.06 ± 0.55 a |
| T5 | 15.05 ± 0.05 ab | 52.91 ± 2.95 ab | 9.02 ± 0.19 a | 37.60 ± 3.59 de | 5.46 ± 0.01 c | 3.89 ± 1.00 cd |
| T6 | 14.33 ± 0.08 bc | 43.26 ± 0.89 cd | 8.22 ± 1.15 a | 56.42 ± 2.24 b | 4.59 ± 0.01 f | 6.11 ± 0.55 ab |
| T7 | 13.81 ± 0.75 bc | 51.41 ± 1.33 ab | 8.93 ± 2.11 a | 42.47 ± 2.29 c | 5.50 ± 0.01 b | 6.74 ± 1.00 a |
| T8 | 12.51 ± 0.46 d | 47.08 ± 3.06 bc | 7.88 ± 2.09 a | 40.54 ± 2.83 cd | 4.80 ± 0.01 e | 4.84 ± 1.00 bcd |
| *p*-value | 0.000 | 0.000 | 0.001 | 0.000 | 0.000 | 0.001 |

T1: commercial peat moss (Control); T2: coconut coir dust (CD); T3: leaf compost (LC); T4: food-waste compost (FC), T5: CD:LC = 1:1 *v*/v, T6: CD:FC = 1:1 *v*/*v*, T7: LC:FC = 1:1 *v*/*v*, T8: CD:LC:FC = 1:1:1 *v*/*v*). According to Duncan's multiple-range test at $p \leq 0.05$, the mean with different letters in the same column indicates a significant difference.

### 3.2. Germination Index

The lowest recorded germination rate of sunflower and water-spinach microgreens was related to 100% food-waste compost used as the growing media, which showed a reduction for the same microgreens by 22.41% and 44.82%, respectively, as compared to the control. A noticeable aspect was a significant decrease in the germination rate of sunflower microgreens in proportion to the utilization of food-waste compost of more than 50% *v*/*v*, a tendency not observed in water-spinach microgreens. In terms of sunflower microgreens, the level of germination index, an estimate of the percentage and speed of germination, increased under the treatment of 1:1 *v*/*v* of coconut coir dust + leaf compost (32.08) and coconut coir dust (31.56), compared to the control, which showed no significant difference. However, the highest mean germination time (longest to germinate) was related to the 100% food-waste compost used as the growing media. For the water-spinach microgreens, all growing media except the food-waste compost gave a higher percent germination, 74.67–90.67%, without any significant difference from the control (Table 3).

**Table 3.** Germination test of sunflower and water-spinach microgreens under different conditions of growing media.

| Treatment | Sunflower | | | Water Spinach | | |
|---|---|---|---|---|---|---|
| | Germination Rate (%) | Germination Index | Mean Germination Time (Day) | Germination Rate (%) | Germination Index | Mean Germination Time (Day) |
| T1 (Control) | 77.33 ± 2.31 ab | 29.12 ± 1.00 a | 2.83 ± 0.16 d | 77.33 ± 4.62 a | 16.47 ± 2.13 c | 5.24 ± 0.73 b |
| T2 | 80.00 ± 6.93 ab | 31.56 ± 3.42 a | 2.75 ± 0.06 d | 90.67 ± 16.17 a | 29.65 ± 5.06 a | 3.46 ± 0.14 c |
| T3 | 81.33 ± 14.05 ab | 20.72 ± 4.16 b | 4.13 ± 0.29 bc | 76.00 ± 6.93 a | 21.95 ± 3.11 b | 4.03 ± 0.72 c |
| T4 | 60.00 ± 8.00 c | 11.05 ± 2.27 c | 5.89 ± 0.97 a | 42.67 ± 7.57 b | 7.11 ± 0.93 d | 6.73 ± 0.68 a |
| T5 | 89.33 ± 2.31 a | 32.08 ± 3.35 a | 3.38 ± 0.54 cd | 82.67 ± 5.03 a | 25.51 ± 2.43 ab | 3.93 ± 0.92 c |
| T6 | 68.00 ± 6.93 bc | 18.38 ± 1.16 b | 4.16 ± 0.56 b | 78.67 ± 9.24 a | 22.70 ± 0.80 b | 3.92 ± 0.18 c |
| T7 | 68.00 ± 10.58 bc | 16.39 ± 4.43 bc | 4.51 ± 0.50 bc | 77.33 ± 8.33 a | 24.60 ± 3.72 ab | 3.49 ± 0.23 c |
| T8 | 81.33 ± 12.86 ab | 22.53 ± 4.73 b | 3.97 ± 0.54 bc | 74.67 ± 15.14 a | 14.80 ± 2.86 c | 5.38 ± 0.26 b |
| *p*-value | 0.021 | 0.000 | 0.000 | 0.002 | 0.000 | 0.000 |

T1: commercial peat moss (Control); T2: coconut coir dust (CD); T3: leaf compost (LC); T4: food waste compost (FC), T5: CD:LC = 1:1 *v*/*v*, T6: CD:FC = 1:1 *v*/*v*, T7: LC:FC = 1:1 *v*/*v*, T8: CD:LC:FC = 1:1:1 *v*/*v*). According to Duncan's multiple-range test at $p \leq 0.05$, the mean with different letters in the same column indicates a significant difference.

### 3.3. Yield Characteristics

The growing media significantly affected microgreen morphological characteristics and all indices of yield components (Figure 1, Tables 4 and 5). For sunflower microgreens, the highest recorded fresh-shoot weight was found for the treatment of 1:1 *v*/*v* of coconut

coir dust + leaf compost (10,114.81 g m$^{-2}$), which was up to 16% higher than the control treatment (peat moss). However, using food waste as the growing media caused a problem with sunflower microgreens, which the fresh marketable weight (the cotyledon leaves had developed with one set of true leaves, approximately 5–10 cm tall) could not sustain from 100% of food-waste compost media, meaning that there was 100% of waste yield. Sunflower microgreens' lowest shoot water content was observed at 100% food-waste compost media. The traits increased with a decreased food-waste compost ratio in the growing media. In addition, the dry shoot matter was significantly highest under the treatment of 100% of food-waste compost media, whereas sunflower microgreens grown under the non-mixed food-waste compost in the growing media showed lower shoot dry matter (3.00–4.55%).

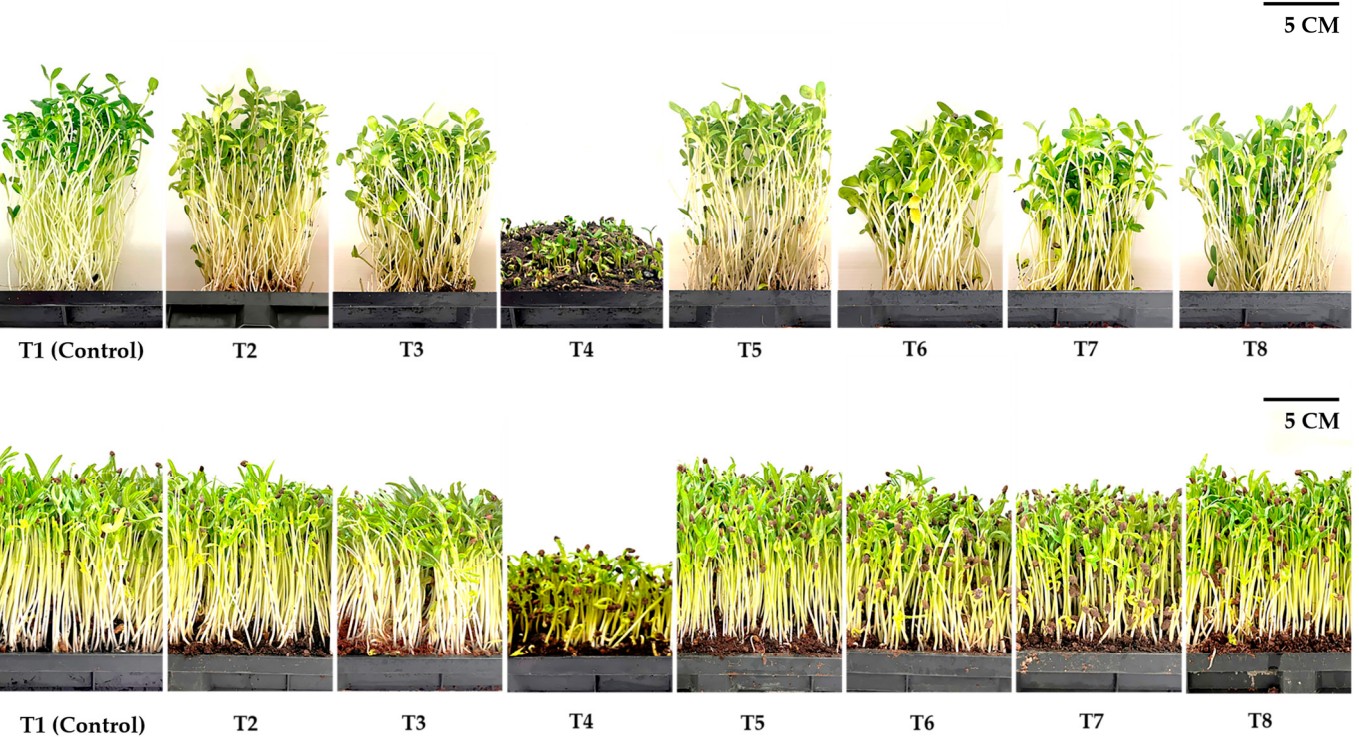

**Figure 1.** Growth characteristics of sunflower and water-spinach microgreens under different growing media (T1: commercial peat moss (Control), T2: coconut coir dust (CD), T3: leaf compost (LC), T4: food-waste compost (FC), T5: CD: LC = 1:1 *v/v*, T6: CD:FC = 1:1 *v/v*, T7: LC:FC = 1:1 *v/v*, T8: CD:LC:FC = 1:1:1 *v/v*).

**Table 4.** Yield characteristics of sunflower microgreens under different growing media.

| Treatment | Shoot Fresh Weight (g m$^{-2}$) | Shoot Dry Weight (g m$^{-2}$) | Non-Marketable Yield (g m$^{-2}$) | Waste Yield Percentage (%) | Shoot Water Content (%) | Shoot Dry Matter (%) |
|---|---|---|---|---|---|---|
| T1 (Control) | 8714.8 ± 146.7 b | 342.9 ± 12.6 a | 614.8 ± 51.3 c | 6.58 ± 0.42 e | 96.06 ± 0.09 ab | 3.94 ± 0.09 de |
| T2 | 8440.7 ± 555.3 b | 307.7 ± 7.0 ab | 666.7 ± 163.7 c | 7.33 ± 1.86 e | 96.34 ± 0.28 ab | 3.66 ± 0.28 de |
| T3 | 6837.0 ± 702.4 c | 309.7 ± 30.5 ab | 974.1 ± 115.7 b | 12.51 ± 1.60 de | 95.45 ± 0.47 ab | 4.55 ± 0.47 cde |
| T4 | 0.0 ± 0.00 f | 0.0 ± 0.0 d | 251.9 ± 6.4 d | 100.00 ± 0.00 a | 75.68 ± 5.00 c | 24.32 ± 1.92 a |
| T5 | 10,114.8 ± 734.5 a | 303.4 ± 35.3 ab | 666.7 ± 80.1 c | 6.23 ± 1.11 e | 97.00 ± 0.27 a | 3.00 ± 0.27 e |
| T6 | 3911.1 ± 416.8 de | 225.6 ± 15.2 c | 1044.4 ± 72.9 ab | 21.12 ± 0.70 c | 94.20 ± 0.53 ab | 5.80 ± 0.53 c |
| T7 | 3344.5 ± 147.3 e | 230.4 ± 68.2 c | 1174.1 ± 100.2 a | 27.86 ± 9.39 b | 92.24 ± 4.27 b | 7.76 ± 0.85 b |
| T8 | 4988.9 ± 100.8 d | 242.3 ± 65.1 bc | 770.4 ± 67.0 c | 13.54 ± 1.31 d | 95.17 ± 0.78 ab | 4.83 ± 0.78 cd |
| *p*-value | 0.000 | 0.000 | 0.000 | 0.000 | 0.000 | 0.000 |

T1: commercial peat moss (Control); T2: coconut coir dust (CD); T3: leaf compost (LC); T4: food waste compost (FC), T5: CD:LC = 1:1 *v/v*, T6: CD:FC = 1:1 *v/v*, T7: LC:FC = 1:1 *v/v*, T8: CD:LC:FC = 1:1:1 *v/v*. According to Duncan's multiple-range test at $p \leq 0.05$, the mean with different letters in the same column indicates a significant difference.

**Table 5.** Yield characteristics of water-spinach microgreens under different growing media.

| Treatment | Shoot Fresh Weight (g m$^{-2}$) | Shoot Dry Weight (g m$^{-2}$) | Non-Marketable Yield (g m$^{-2}$) | Waste Yield Percentage (%) | Shoot Water Content (%) | Shoot Dry Matter (%) |
|---|---|---|---|---|---|---|
| T1 (Control) | 10,855.6 ± 211.1 a | 358.8 ± 28.8 a | 966.7 ± 111.1 ab | 8.18 ± 0.03 c | 96.69 ± 0.11 a | 3.31 ± 0.11 c |
| T2 | 10,966.7 ± 266.7 a | 343.4 ± 45.6 ab | 844.4 ± 66.7 ab | 7.28 ± 1.10 c | 96.87 ± 0.06 a | 3.13 ± 0.06 c |
| T3 | 8655.6 ± 77.8 b | 317.4 ± 24.1 ab | 922.2 ± 88.9 ab | 9.62 ± 0.77 bc | 96.33 ± 0.25 ab | 3.67 ± 0.25 bc |
| T4 | 72.2 ± 5.6 d | 4.5 ± 0.3 c | 500.0 ± 44.4 c | 87.30 ± 1.84 a | 93.72 ± 0.87 c | 6.28 ± 0.87 a |
| T5 | 10,059.3 ± 50.6 ab | 354.8 ± 30.4 a | 833.3 ± 131.0 b | 7.59 ± 1.13 c | 96.48 ± 0.13 ab | 3.52 ± 0.13 bc |
| T6 | 6633.3 ± 1628.9 c | 270.8 ± 89.8 b | 559.3 ± 44.9 c | 8.14 ± 0.96 c | 95.97 ± 0.37 b | 4.03 ± 0.37 b |
| T7 | 6533.3 ± 742.0 c | 269.6 ± 21.1 b | 933.3 ± 77.8 ab | 12.25 ± 2.92 b | 95.86 ± 0.21 b | 4.14 ± 0.21 b |
| T8 | 9800.0 ± 423.5 ab | 328.7 ± 15.5 ab | 1000.0 ± 80.1 a | 9.28 ± 2.58 bc | 96.65 ± 0.03 a | 3.35 ± 0.03 c |
| *p*-value | 0.000 | 0.000 | 0.000 | 0.000 | 0.000 | 0.000 |

T1: commercial peat moss (Control); T2: coconut coir dust (CD); T3: leaf compost (LC); T4: food waste compost (FC), T5: CD:LC = 1:1 *v/v*, T6: CD:FC = 1:1 *v/v*, T7: LC:FC = 1:1 *v/v*, T8: CD:LC:FC = 1:1:1 *v/v*). According to Duncan's multiple-range test at $p \leq 0.05$, the mean with different letters in the same column indicates a significant difference.

Regarding water-spinach microgreens, the impact of growing media treatments on yield characteristics was significant at a 1% level. The highest fresh-shoot weight was recorded under the treatments of coconut coir dust, peat moss (control), 1:1 *v/v* of coconut coir dust + leaf compost, and 1:1:1 *v/v* of coconut coir dust + leaf compost + food-waste compost, which had an average fresh-shoot weight between 10,966.67–9800.00 g m$^{-2}$. However, the lowest fresh-shoot weight or marketable yield was still obtained for the 100% of food-waste compost media (72.22 g m$^{-2}$), with an 87.30% waste-yield percentage observed. In addition, growing media without food-waste compost or with food-waste compost less than 50% of all raw materials caused an increase in shoot water content (96.48–96.87%) as well as decreased the dry-shoot matter (3.13–3.67%) compared to 100% of food-waste compost media (Table 5).

### 3.4. Phytochemical Composition and Nitrate Accumulation

The ANOVA results demonstrated that variations in growing media had an influence ($p \leq 0.05$) on some biochemical compositions of both microgreens, except the total phenolic content, flavonoid content, and DPPH radical scavenging of sunflower microgreens (Tables 6 and 7). In terms of sunflower microgreens, the total chlorophyll, carotenoid, and flavonoid content in plant tissues were significantly highest when cultivated in 1:1 *v/v* of coconut coir dust + leaf-compost growing media, in which the total chlorophyll and carotenoid content displayed no significant difference from the control (peat moss). A significant variation in the total phenolic content, flavonoid content, and DPPH radical scavenging of sunflower microgreens was not observed, with a range from 9.44–11.48 mg GAE g$^{-1}$ DW, 24.91–27.66 mg g$^{-1}$ DW, and 41.59–44.69%, respectively. However, the highest nitrate content in sunflower microgreens was obtained in the leaf-compost media (1283.01 mg kg$^{-1}$ FW), which showed no significant difference from sunflower microgreens that were cultivated in 1:1 *v/v* of coconut coir dust + leaf compost (1166.18 g kg$^{-1}$ FW) (Table 6).

Concerning the biochemical composition results of water spinach microgreens, the total chlorophyll, carotenoid, and total phenolic contents in plant tissues were significantly higher when cultivated in a growing media consisting of coconut coir dust or 1:1 *v/v* of coconut coir dust + leaf compost, with no significant difference from that of the control (peat moss). However, the flavonoid content in water-spinach microgreens cultivated in 1:1 *v/v* of coconut coir dust + leaf-compost growing media was nearly 4% higher than in the control treatment. The lowest DPPH radical scavenging of water-spinach microgreens was found for the 100% food-waste compost media (36.13%). In comparison, the other treatments showed DPPH radical scavenging detected in microgreens ranging from 42.11 to 44.82%. Likewise, a noticeable aspect was that the nitrate content in water-spinach microgreens was significantly highest when they were cultivated in a growing media consisting of 1:1

$v/v$ of coconut coir dust + leaf compost (1009.61 mg kg$^{-1}$ FW), which was around 2 times more than that of the control treatment (Table 7).

**Table 6.** Yield quality and nitrate content of sunflower microgreens under different growing media.

| Treatment | Total Chlorophyll (mg 100 g$^{-1}$ FW) | Carotenoid (mg 100 g$^{-1}$ FW) | Total Phenolic (mg GAE g$^{-1}$ DW) | Flavonoid (mg g$^{-1}$ DW) | DPPH Radical Scavenging (%) | Nitrate Content (mg kg$^{-1}$ FW) |
|---|---|---|---|---|---|---|
| T1 (Control) | 17.76 ± 0.89 a | 194.4 ± 12.0 a | 11.43 ± 1.07 | 25.63 ± 0.06 | 41.73 ± 1.95 | 495.8 ± 48.3 cd |
| T2 | 13.79 ± 2.12 b | 146.8 ± 12.7 b | 11.09 ± 0.44 | 25.51 ± 0.41 | 41.59 ± 0.68 | 588.8 ± 46.4 bc |
| T3 | 13.48 ± 1.15 b | 135.6 ± 28.8 b | 11.48 ± 0.68 | 24.91 ± 0.18 | 42.67 ± 1.88 | 1283.0 ± 171.3 a |
| T4 | ND | ND | ND | ND | ND | ND |
| T5 | 18.69 ± 1.24 a | 187.6 ± 10.4 a | 11.06 ± 0.63 | 26.83 ± 1.51 | 43.71 ± 2.38 | 1166.2 ± 13.9 a |
| T6 | 4.93 ± 0.45 d | 49.0 ± 5.5 c | 9.44 ± 0.19 | 27.66 ± 2.58 | 42.68 ± 0.29 | 505.4 ± 31.1 cd |
| T7 | 4.81 ± 0.26 d | 54.7 ± 5.1 c | 10.25 ± 1.63 | 27.58 ± 1.07 | 44.13 ± 2.26 | 377.3 ± 41.9 d |
| T8 | 7.05 ± 1.01 c | 66.2 ± 7.2 c | 10.41 ± 0.57 | 27.09 ± 1.21 | 44.69 ± 2.38 | 676.4 ± 104.6 b |
| *p*-value | 0.000 | 0.000 | 0.103 | 0.112 | 0.357 | 0.000 |

T1: commercial peat moss (Control); T2: coconut coir dust (CD); T3: leaf compost (LC); T4: food- waste compost (FC), T5: CD:LC = 1:1 $v/v$, T6: CD:FC = 1:1 $v/v$, T7: LC:FC = 1:1 $v/v$, T8: CD:LC:FC = 1:1:1 $v/v$). According to Duncan's multiple-range test at $p \leq 0.05$, the mean with different letters in the same column indicates a significant difference. ND: no sample detected.

**Table 7.** Yield quality and nitrate content of water-spinach microgreens under different growing media.

| Treatment | Total Chlorophyll (mg 100 g$^{-1}$ FW) | Carotenoid (mg 100 g$^{-1}$ FW) | Total Phenolic (mg GAE g$^{-1}$ DW) | Flavonoid (mg g$^{-1}$ DW) | DPPH Radical Scavenging (%) | Nitrate Content (mg kg$^{-1}$ FW) |
|---|---|---|---|---|---|---|
| T1 (Control) | 32.54 ± 2.20 a | 401.2 ± 53.7 a | 12.47 ± 0.65 ab | 24.33 ± 0.26 bcd | 43.72 ± 1.13 a | 527.3 ± 33.3 de |
| T2 | 32.79 ± 5.80 a | 386.6 ± 44.3 a | 13.72 ± 0.74 a | 24.50 ± 0.76 bcd | 42.80 ± 1.64 a | 534.4 ± 13.8 d |
| T3 | 22.20 ± 0.40 b | 283.7 ± 15.2 b | 12.44 ± 0.98 ab | 24.64 ± 0.10 bc | 44.66 ± 0.55 a | 826.3 ± 24.8 b |
| T4 | 0.08 ± 0.01 e | 0.9 ± 0.1 d | 11.34 ± 1.74 bc | 24.92 ± 0.77 ab | 36.13 ± 3.20 b | 478.8 ± 22.3 ef |
| T5 | 30.12 ± 4.10 a | 359.3 ± 38.8 a | 11.63 ± 1.30 abc | 25.56 ± 0.69 a | 43.29 ± 1.91 a | 1009.6 ± 56.6 a |
| T6 | 8.74 ± 0.95 c | 113.8 ± 11.8 c | 11.57 ± 2.00 abc | 24.08 ± 0.13 bcd | 44.82 ± 0.72 a | 553.2 ± 11.2 d |
| T7 | 7.66 ± 0.65 c | 90.7 ± 5.7 c | 9.82 ± 0.53 cd | 23.85 ± 0.24 cd | 42.55 ± 1.08 a | 472.3 ± 32.5 f |
| T8 | 20.43 ± 2.60 b | 249.0 ± 23.6 b | 8.89 ± 0.81 d | 23.63 ± 0.30 d | 42.11 ± 0.43 a | 647.0 ± 23.2 c |
| *p*-value | 0.000 | 0.000 | 0.004 | 0.004 | 0.000 | 0.000 |

T1: commercial peat moss (Control); T2: coconut coir dust (CD); T3: leaf compost (LC); T4: food waste compost (FC), T5: CD:LC = 1:1 $v/v$, T6: CD:FC = 1:1 $v/v$, T7: LC:FC = 1:1 $v/v$, T8: CD:LC:FC = 1:1:1 $v/v$). According to Duncan's multiple range test at $p \leq 0.05$, the mean with different letters in the same column indicates a significant difference.

### 3.5. Microbial Populations on Microgreens

Concerning microbiological contamination of all microgreen traits, the preliminary microbiological analyses revealed that all growing media did not cause contamination by pathogenic microorganisms in sunflower and water spinach microgreens; namely, *Clostridium perfringens*, *Salmonella* spp., and *Staphylococcus aureus* did not exceed the legal limits recommended by Thai agricultural standards (TAS 9007-2005: Safety requirements for agricultural commodity and food) (Tables 8 and 9). However, almost all growing media resulted in a higher population of *Bacillus cereus* contamination in both microgreens than the standard set limit ($\leq 5.0 \times 10^2$ CFU g$^{-1}$). A higher population of *Bacillus cereus* was observed in both microgreens when cultivated in a growing media consisting of 100% food-waste compost or 50% mixed in the media than in peat moss (control), coconut coir dust, or leaf compost. In addition, only sunflower microgreens grown in peat moss and coconut coir dust had a population of *Bacillus cereus* within the standard limit (Table 8).

**Table 8.** Quantitative data of pathogenic bacteria in sunflower microgreens under different growing media.

| Treatment | *Bacillus cereus* (CFU g$^{-1}$) | *Clostridium perfringens* (CFU g$^{-1}$) | *Salmonella* spp. (CFU g$^{-1}$) | *Staphylococcus aureus* (CFU g$^{-1}$) |
|---|---|---|---|---|
| T1 | $2.8 \times 10^2$ | <10 | Non-detected | <10 |
| T2 | $1.0 \times 10^2$ | <10 | Non-detected | <10 |
| T3 | $4.2 \times 10^3$ | <10 | Non-detected | <10 |
| T4 | $7.0 \times 10^3$ | <10 | Non-detected | <10 |
| T5 | $2.6 \times 10^3$ | <10 | Non-detected | <10 |
| T6 | $1.2 \times 10^4$ | <10 | Non-detected | <10 |
| T7 | $1.2 \times 10^4$ | <10 | Non-detected | <10 |
| T8 | $4.3 \times 10^3$ | <10 | Non-detected | <10 |
| Thailand standard * | $\leq 5.0 \times 10^2$ CFU g$^{-1}$ | $\leq 1.0 \times 10^2$ CFU g$^{-1}$ | Non-detected in 25 g | $\leq 1.0 \times 10^2$ CFU g$^{-1}$ |

T1: commercial peat moss (Control); T2: coconut coir dust (CD); T3: leaf compost (LC); T4: food waste compost (FC), T5: CD:LC = 1:1 *v/v*, T6: CD:FC = 1:1 *v/v*, T7: LC:FC = 1:1 *v/v*, T8: CD:LC:FC = 1:1:1 *v/v*). * Thai agricultural standard TAS 9007-2005. Safety requirements for agricultural commodities and food.

**Table 9.** Quantitative data of pathogenic bacteria in water-spinach microgreens under different growing media.

| Treatment | *Bacillus cereus* (CFU g$^{-1}$) | *Clostridium perfringens* (CFU g$^{-1}$) | *Salmonella* spp. (CFU g$^{-1}$) | *Staphylococcus aureus* (CFU g$^{-1}$) |
|---|---|---|---|---|
| T1 | $5.8 \times 10^2$ | <10 | Non-detected | <10 |
| T2 | $1.2 \times 10^3$ | <10 | Non-detected | <10 |
| T3 | $6.0 \times 10^3$ | <10 | Non-detected | <10 |
| T4 | $1.8 \times 10^4$ | <10 | Non-detected | <10 |
| T5 | $4.5 \times 10^3$ | <10 | Non-detected | <10 |
| T6 | $5.7 \times 10^3$ | <10 | Non-detected | <10 |
| T7 | $6.2 \times 10^3$ | <10 | Non-detected | <10 |
| T8 | $5.6 \times 10^3$ | <10 | Non-detected | <10 |
| Thailand standard * | $\leq 5.0 \times 10^2$ CFU g$^{-1}$ | $\leq 1.0 \times 10^2$ CFU g$^{-1}$ | Non-detected in 25 g | $\leq 1.0 \times 10^2$ CFU g$^{-1}$ |

T1: commercial peat moss (Control); T2: coconut coir dust (CD); T3: leaf compost (LC); T4: food waste compost (FC), T5: CD:LC = 1:1 *v/v*, T6: CD:FC = 1:1 *v/v*, T7: LC:FC = 1:1 *v/v*, T8: CD:LC:FC = 1:1:1 *v/v*). * Thai agricultural standard TAS 9007-2005. Safety requirements for agricultural commodities and food.

## 4. Discussion

Several factors directly affect the yield and quality of microgreens, including the microenvironment (light spectrum, temperature, humidity, etc.), growth media, fertilization, and pre- and postharvest treatments (pre-sowing seed soaking, postharvest UV-B treatment, etc.) [7,30,31]. A high yield has been considered the leading indicator for microgreen production, in which the growing media plays an essential factor in the growth and yield. Previous literature has reported alternative growing media that could be used instead of the standard peat-based media. Attempts have been made to develop media derived from industrial or agricultural wastes that are environmentally friendly, low-cost, easily disposable, and potentially reusable, while also being comparable to peat-based media in terms of resulting yields [11–13]. However, the physicochemical properties of the growing media play a significant role in determining the yield and quality of microgreens, whereby the ideal growing media should have an adequate ratio of micropores and macropores, satisfactory pH (5.5–6.5), high water retention capacity, and sufficient nutrients [14,18,32]. In the current experiment, almost all growing media had pH values within the optimal range (5.50–6.59), whereas the growing media consisting of food-waste compost had average pH values slightly higher than the ideal maximum values, which were in a range from 6.68–6.82. In addition, 100% food-waste compost media also had the highest EC and Na

concentration (Tables 1 and 2), which might have affected the lower germination rate of both microgreens (Table 3). These results were similar to those of a previous study by Kang et al. [18], who indicated that fertilization by using food waste might have a negative effect on seedlings' growth via salinity stress, namely excess Na concentration, which is attributed to ionic imbalance and leads to inhibiting plant growth. Although the 100% leaf-compost media had significantly higher N and Mg, the growth and yield of both microgreens were lower, and in the 50% leaf compost mixed with coconut coir dust (1:1 $v/v$ of CD:LC), the significant differences of P, K, and Ca concentrations were not observed. Generally, it is well-known that nutrient exposures could lead to changes in metabolic seed mechanisms, which may involve seed-nutrient balance and physiological seed deterioration [33,34]. These mechanisms need to be explored in future studies to find each nutrient's influence on the germination process of microgreens. However, our findings demonstrated that lower bulk density and higher total pore-space values of the growing media resulted in the higher yield of both microgreens (Tables 1, 4 and 5). The bulk-density values of the growing media indicate the porosity and water-buffering capacity [12,32], whereby higher bulk density could result in plant-growth limitation and increased transportation costs, which might not be desirable for growers during planting preparation. Although it is difficult to recommend the adequate bulk density of growing media for microgreen production depending on growing and irrigation techniques, the suitable bulk-density values (67.64–263.62 kg m$^{-3}$) of the growing media for the production of both microgreens in the current study were near the ideal range suggested by Fernandes and Corá [32], who reported that values of bulk density lower than or nearly 300 g L$^{-1}$ (kg m$^{-3}$) are considered acceptable for seedling propagation.

Chlorophyll and carotenoid are classified as plant pigments that can absorb light energy and respond to transform it into chemical energy in the photosynthesis process [35]. In addition, both pigments may function as secondary metabolites that display more potential nutritional and health benefits due to their antioxidant functions, such as being free radical scavengers, encouraging eye and bone health, and reducing cancer risk [36,37]. In the present results, chlorophyll and carotenoid contents tended to follow the growth and yield of both microgreens under different growing media. The obtained results for both microgreens grown on 1:1 $v/v$ of coconut coir dust + leaf compost or coconut coir dust (for only water-spinach microgreens) look promising for the creation of a product with a high nutraceutical value of plant pigments, similar to the performance of commercial growing media such as peat moss. However, the change in the other phytochemical compounds, such as total phenolic, flavonoid, and DPPH radical scavenging of both microgreens under different growing media was complicated because microgreens are mainly composed of single shoots with cotyledon leaves. There is evidence that the concentration of chlorophylls and carotenoids is lower than that of mature leaves or the adult stage of the same species as a result of the physiological responsibility, and some literature could not find a significant effect of growing media on phytochemical compounds with a short harvesting cycle [13,38]. Furthermore, it is interesting to note that the total chlorophyll and carotenoid of both microgreens grown in 100% food-waste media half-mixed with other growing media were lower than those of the other growing media, which may be attributed to excess Na ions interfering with K or Ca uptake in the plant tissues, resulting in chlorosis [39]. In addition, it also remains to be explained whether direct (elemental) or indirect (lipid peroxidation, membrane stability, etc.) damage to cellular components may occur due to Na ions.

High levels of nitrate residue in microgreens are considered to have possible adverse effects on human health that could be controlled by decreasing the N fertilization rate [40]. However, levels of nitrate accumulation in both microgreens grown under different growing media (377.29–1283.01 mg NO$_3^-$ kg$^{-1}$ FW) did not exceed the legal limit set by the European Union under Commission Regulation (EC) No. 1881/2006, which establishes the maximal nitrate levels of fresh spinach and lettuce between 2000–5000 mg NO$_3^-$ kg$^{-1}$ fresh weight. Data related to lower nitrate accumulation in microgreens than that in baby greens or adult plants of the same species have been reported in several studies [13,41,42]. This is

probably due to higher nitrogen absorption in plant tissues via increasing the plant-growth stage, especially vegetative and early reproductive stages, for biomass production [43].

Microgreens are typically consumed raw; all pathogens causing foodborne outbreaks associated with these products are considered potential hazards for microgreen risk assessment. Contamination via foodborne pathogens in microgreens may occur at the outset from seed, growing media, irrigation water, workers' hygiene, and equipment [44,45]. Additional studies also observed that cultivation materials could cause contamination of microgreens, even with indoor or soilless culture in hydroponics systems, by providing nutrient and humidity sources [46]. Similarly, Işık et al. [15] also reported that transferring *E. coli* from contaminated growth media, such as peat moss and perlite, to the edible part of microgreens is possible. However, the current study demonstrated that all growing media did not cause pathogenic contamination in both microgreens, namely *C. perfringens*, *Salmonella* spp., and *S. aureus*. While only sunflower microgreens grown in peat moss and coconut coir dust had a population of *B. cereus* within the standard limit ($5 \times 10^2$ CFU g$^{-1}$), almost all the growing media resulted in a higher population of *B. cereus* in both microgreens than the standard set limit. *B. cereus*, a gram-positive spore-forming bacterium, can be found in various natural sources, including soil and water, and this pathogen contamination in fresh-cut vegetables or microgreens may occur from cultivation or processing lines [47]. Although *B. cereus* is often found as a contaminant in fresh vegetables and can cause emetic or diarrheal syndromes, it is generally thought that most cases of foodborne outbreaks caused by the *B. cereus* group have been associated with concentrations above $10^5$ CFU g$^{-1}$ food [48,49]. However, to prevent or reduce the foodborne incidences caused by *B. cereus* in microgreens, the produce should be cleaned at least two times with water. In addition, they should not be stored at a temperature (5–60 °C) favorable to *B. cereus* growth but be kept in a refrigerator at 4 °C or lower if they are not meant for immediate consumption [50]. Although conventional surface sanitation methods could reduce microbial contamination in microgreens, the pathogens would not be eliminated if present and commonly would start to contaminate from the raw materials [44]. The results of this study further confirm that alternative growing media made from agricultural wastes, such as coconut coir dust or coconut coir dust mixed with leaf compost (1:1 *v/v*), could be utilized for water spinach and sunflower microgreen production. However, for widely upscale commercial production, agricultural waste as a growing media should first be sterilized to protect the plants from microbial contamination in the first step of microgreen production.

## 5. Conclusions

This research further demonstrated that growing media with beneficial physicochemical properties significantly affect the microgreens' yield and quality, including microbial contamination. Overall, this study provides valuable data for determining the growing media discarded from agricultural processing that could be used as a low-cost and renewable alternative to peat moss to produce sunflower and water spinach microgreens. A significantly higher yield of sunflower microgreens was observed when they were cultivated in 1:1 *v/v* of coconut coir dust + leaf compost medium instead of the control media (peat moss). In comparison, a higher yield of water-spinach microgreens was recorded under the treatments of coconut coir dust, 1:1 *v/v* of coconut coir dust + leaf compost, and 1:1:1 *v/v* of coconut coir dust + leaf compost + food-waste compost, without any significant difference from the control (peat moss). In addition, under these growing media conditions in each microgreen's production, a higher yield quality in total chlorophyll and carotenoid was observed, and the nitrate residue did not exceed the standard limit set by the EU. Contrarily, utilizing a higher ratio (50–100%) of food-waste compost mixed into the growing media resulted in the growth inhibition of microgreens, which may be caused by the excess salt content in food waste. In addition, all growing media did not cause pathogenic contamination in both microgreens, namely *C. perfringens*, *Salmonella* spp., and *S. aureus*. However, almost all growing media resulted in a higher population of *B. cereus* contamination in both microgreens than the standard set limit; therefore, further cleaning

before consumption is recommended to avoid or reduce the foodborne incidences caused by *B. cereus* in microgreens.

**Author Contributions:** Conceptualization, O.T.; methodology, O.T.; formal analysis, O.T.; investigation, O.T., N.S. and N.P.; resources, O.T.; data curation, O.T.; writing—original draft preparation, O.T.; writing—review and editing, O.T., P.C. (Preuk Chutimanukul), D.A., W.C., P.P., V.V. and P.C. (Panita Chutimanukul); supervision, H.E.; funding acquisition, O.T. All authors have read and agreed to the published version of the manuscript.

**Funding:** This research was funded by Thammasat University Research Fund, Contract No. TUFF 06/2565.

**Data Availability Statement:** Data are contained within the article.

**Acknowledgments:** We thank the Thammasat University Center of Excellence in Agriculture Innovation Center through the Supply Chain and Value Chain and the Department of Agricultural Technology, Faculty of Science and Technology, Thammasat University for providing experimental and laboratory facilities.

**Conflicts of Interest:** The authors declare no conflict of interest. The funders had no role in the design of the study; in the collection, analyses, or interpretation of data; in the writing of the manuscript; or in the decision to publish the results.

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
