# Peer review of "Efficacy of Agricultural and Food Wastes as the Growing Media for Sunflower and Water Spinach Microgreens Production"

_horticulturae, doi:10.3390/horticulturae9080876_

Round 1

Reviewer 1 Report

The paper entitled ‘Efficacy of agricultural and food wastes as the growing media for sunflower and water spinach microgreens production’ explores the possibility to use agriculture and food wastes based growing media for microgreens cultivation, especially sunflower and spinach. Authors have provided with data sufficient evidence on substrate quality with respect to texture, contamination with diseases. I report some suggestions in order to improve the quality of manuscript, which are as follow:

L 25-26: Mention if the change in biochemical composition of microgreens due to growing media, and highlight the most prominent change or trend to attract reader to read the full paper.

L 30: Conclude from the study which growing media is useful and safe?

L 52: hence becomes (resulting in) quite expensive.

L 84: It is better o write eight ‘different’ growing media, rather eight ‘difference in’…

L 92: The leaf wastes are were placed

L 98: … better write in past tense as you did it by your own: ‘the composting was assumed to be finished when the pile was no longer heating up’

L 191: do you mean shoot water content (%) on the right side of equal sign?

In Table 4 & 5 restrict data of Shoot fresh weight, Shoot dry weight and Non-marketable yield to single value after decimal.

Same in Table 6 & 7 for Carotenoid and Nitrate content.

L 478: Delete ‘by’

L 520-21: In addition, direct (elemental) or indirect (lipid peroxidation, membrane stability, etc.) damage to cellular components may occur due to Na ion shall also be explained.

L 571: which may cause by ‘saline stress’. Its misleading. It is better to supplement for e.g., excess of salt content in food wastes.

Its ok, only some minor correction is needed.

Reviewer 2 Report

This paper emphasized on the cultivation of microgreens in wastes. In general, this paper is well-written and the results are interesting. In my view, this research will be helpful to the production of microgreens in modern agriculture. Minor revision is needed before the acceptance of this manuscript.

(1) Line 113, sunflower (Helianthus annuus L.) and water spinach (Ipomoea aquatica Forsk.) were selected for this experiment. Can you provide the reasons for the selection of these two microgreens?   

(2) Line 460, a paragraph should be added to discuss the advantages of employing agricultural and food wastes for microgreens cultivation.  

(3) Line 550, some “pathogenic bacteria” were detected by the researchers in this study. Please briefly discuss the methods to control these “pathogenic bacteria”. In a real-world application, otherwise, “pathogenic bacteria” may threaten the consumers health.    

(4) Line 555, this study studied the feasibility of growing microgreens in wastes. In the practical application, however, this novel agricultural model has not been widely adopted. So please discuss how to promote the wide application of this novel agricultural model in the coming future.  

Reviewer 3 Report

The peer-reviewed article "Efficacy of agricultural and food wastes as the growing media for sunflower and water spinach microgreens production" is an original scientific article, which the theme has been properly described.

I would like to congratulate Authors for the good-quality of the article, the literature reported used to write the paper, and for the clear and appropriate structure. The manuscript is well written, presented and discussed, and understandable to a specialist readership. However, a few improvements in manuscript are required before the publication.

line 176, please put [24] instead of „(Ibrahim et al., 2019)”

line 436, should be „(Table 8-9)”

line 442, should be „(Table 8)”

Reviewer 4 Report

Dear Authors,

Thank you for the opportunity to contribute to your work as a reviewer. The topic of the research covers a very important topic, which is a possible contribution to enhance the sustainability of growing media in PFAL systems, which is a strong limitation of such systems. The experiment itself has a very complex approach, which supports the reliability of the outcomes.

General comments

I suggest elaborating on the last paragraph of the introduction section. The main focus of the study is the organic materials to be used for growing media, therefore some more information, previous studies have to be introduced here.

section 2.1 contains insufficient information about the media used. A table might help enhancing understandability. Consider using different acronyms for media instead of T as it is quite non-informative. At least, change the letterings for controls. Line 84-87 are very chaotic. How did you determine the ratios of food waste compost? What did you want to model with the applied ratios?

It would be useful to assess the applied media in a complex way at the end of the paper. Some future perspectives are also welcome, which can show the path for developments.

Detailed comments

-          Consider adding one more introductory sentence at the beginning of the abstract. I suggest removing the very general statement about the popularity of microgreens as it is obvious.

-          line 49, sentence starting with: Peat-based: reword. The problem with peat is not its price, but the fact, that it takes a lot of time to renew, therefore it is rather unsustainable.

-          line 57, sentence starting with Moreover: reword.

-          line 94, sentence starting with Stack: reword.

-          Line 118: reword sentence starting here.

-          line 278: Add the name of statistical program used.

-          line 462: UV-B

-          line 465: Sentence starting with Alternative: reword.

Although I am not a native English speaker, I strongly suggest improving the English use and correct the grammatical errors. Several sentences require rewording.

Reviewer 5 Report

Dear Authors, 

Many thanks for the interesting manuscript of using food and agricultural waste as an alternative of growing media for the production of sunflower and water spinach microgreen.

Suggestion:

Change the title to:

"Effectiveness of agricultural and food waste as growing media in sunflower and water spinach microgreen production"

in Materials and methods:

Line 251 - include "microbial analysis"

in Statistical Analysis:

Line 280 - include the information of using a statistical programme or software, please - how did you analyse your data?

The writing is clear and the steps and methods are well explained. The English is readable.
